# Pollution Levels and Risk Assessment of Heavy Metals in the Soil of a Landfill Site: A Case Study in Lhasa, Tibet

**DOI:** 10.3390/ijerph191710704

**Published:** 2022-08-27

**Authors:** Peng Zhou, Dan Zeng, Xutong Wang, Lingyu Tai, Wenwu Zhou, Qiongda Zhuoma, Fawei Lin

**Affiliations:** 1School of Science, Tibet University, Lhasa 850000, China; 2School of Environmental Science and Engineering, Tianjin University, Tianjin 300072, China; 3Key Laboratory of Agro-Forestry Environmental Processes and Ecological Regulation of Hainan Province, College of Ecology & Environment, Hainan University, Haikou 570228, China; 4The People’s Government of Bahe Township in Gongbo’gyamda, Nyingchi 860000, China

**Keywords:** Tibet, landfill site, soil heavy metal pollution, ecological risk, health risk assessment

## Abstract

As an important ecological security barrier in China, the ecological environment of Tibet has aroused widespread concern domestically and overseas. Landfills are a major solid waste treatment approach in Tibet but also cause severe environmental pollution. To date, there are no studies related to the pollution risk of landfills in Tibetan areas. This study investigated the pollution levels, ecological risk, health risk, and possible pollution sources of eight heavy metals in the soils around a landfill site in Lhasa, Tibet. The results indicated that the concentrations of heavy metals in soil were relatively low, only cadmium (Cd), arsenic (As), copper (Cu), chromium (Cr), zinc (Zn), nickel (Ni), and lead (Pb) were 1–2 times higher than the corresponding background value. The values of the single pollution index and geo-accumulation index show that the study area is most seriously polluted by Cd and As. Based on the Nemerow pollution index and the pollution load index, over 83.3% and 8.33% of soil sampling sites had light and moderate contamination levels. According to the results of potential ecological risk evaluation, the potential ecological risk of heavy metals in soil was very low, and only one out of the 72 sampling sites exhibited considerable ecological risk. Cd, As, and mercury (Hg) served as the dominant ecological risk contributors and contributed over 45.0%, 14.1%, and 18% of the ecological risk. The results of the health risk evaluation showed that adults have a higher risk of cancer (1.73 × 10^−5^), while the non-carcinogenic risk for adults was low. Waste disposal activities and construction activities have a significant influence on soil heavy metal concentrations, causing a higher pollution level in the southeast part of the landfill site in Lhasa.

## 1. Introduction

With the acceleration of urbanization and the rapid increase in the population in China, the amount of municipal solid waste (MSW) is increasing rapidly. Relevant data show that the production of MSW in China has increased by 5–10% each year, and will continue to grow for the next 10–20 years [1], and MSW production will reach 409~528 million tons in 2030–2050 [2]. The landfill disposal method of MSW is used widely in China, due to the low cost, and low requirements for disposal and operation [3,4]. According to the economic and technological level, and the compositions of MSW in China, landfills will still be the main treatment method of MSW for a long time in the future [5,6]. However, in recent years, researchers have reported that landfills cause serious problems such as soil heavy metal contamination, water and air pollution, and ecological risks to the surrounding environment [7,8]. Soil heavy metals could lead to agricultural pollution and the decline of soil productivity, thus posing a threat to ecological security, food safety, and human health [9]. Many studies [10,11] have shown that the cancer risk and non-carcinogenic risk due to soil heavy metals exceed the risk control value of the US Environmental Protection Agency (USEPA).

Qinghai-Tibet Plateau is considered as an important ecological security barrier for its fragile ecosystem and sensitive ecological environment. At present, more than 800,000 tons of MSW are produced in Tibet every year. Due to the slow-moving economy, open stacking and open burning are still carried out in Tibet, which cause environmental pollution and ecological imbalance. To meet the requirements from the rapid increase in MSW in Tibet, landfill should be selected and expanded as the disposal method for MSW. However, many studies have also shown that the environmental quality of this “clean slate” in China is alarming [12]. Heavy metals are found in relatively high concentrations in different matrices such as water, sediment, soil, and plants [13,14,15]. However, there are few studies investigating the environmental pollution of landfills in Tibet.

Based on Tibet’s unique natural environment, this study chooses the largest and longest-running domestic waste landfill as the research object and adopted different methods such as the pollution index (*P_i_*), the Nemerow pollution index (*PN*), the geo-accumulation index (*I_geo_*), and health risk to assess the pollution levels, ecological risk in soil, and human health risk. The results of this study will provide a complete, comprehensive, and scientific basis for prevention of heavy metals pollution and human health risk control in Tibet.

## 2. Materials and Methods

### 2.1. Study Area

The sanitary landfill of MSW is located in Nipugou Village, Niedang Township, Lhasa City. The geographic coordinates are 90°57′31″–90°58′37″ E, 29°33′28″−29°34′49″ N. The mountains are in the north and west parts, the hazardous waste treatment center and MSW incinerator are in the east part, and the leachate treatment station/construction waste dump and residential area are in the south of the landfill site, which is 0.8 km from the residential area. In the dry season, the leachate is sprayed onto the landfill site by a sewage pump for recycling. During the rainy season, it is treated using microbial degradation and natural evaporation. The landfill site (phase I) went into service in 2000 and has a capacity of approximately 1.9028 million·m^3^; the design life is 20 years. The landfill site was expanded (phase II) in 2014, with a reservoir capacity of approximately 9.0781 million·m^3^, and is mainly used to treat wastes such as fly ash and slag from incineration plants of MSW.

### 2.2. Sampling and Chemical Analysis

The sampling sites were distributed in a circular pattern (Figure 1). Field sampling was performed from 18 October to 28 November 2018. A total of 72 samples were collected from topsoil, 5 m, 20 m, 50 m, 100 m, and 500 m in the horizontal direction, covering the east, south, west, north, northeast, northwest, southeast, and southwest of the landfill and, among them, to the sampling sites of 5 m and 20 m were added two sites (5 cm and 20 cm in the vertical direction, respectively). The samples were homogenized, stored in sample bags, and air-dried at room temperature, then passed through a 0.075 mm sieve for chemical analysis.

Soil (about 0.5 g) was dissipated by microwave with 11 mL of a 5:5:1 HNO_3_:HF:H_2_O_2_ (*v*/*v*) solution. Soil pH was detected from the supernatants with a water–soil ratio of 2.5:1, using a pH meter (Shanghai INESA Scientific Instrument Co., Shanghai, China). The concentrations of chromium (Cr), lead (Pb), copper (Cu), zinc (Zn), nickel (Ni), and cadmium (Cd) were measured by a flame atomic absorption spectrometer (Z-500, HITACHI, Beijing, China). Determination of mercury (Hg) and arsenic (As) concentrations was realized by an atomic fluorescence spectrometer (AFS-8220, TITAN, Beijing, China).

### 2.3. Contamination Assessment Methods

This article adopts the single pollution index (*P_i_*), the Nemerow pollution index (*PN*), the geo-accumulation index (*I_geo_*), and the pollution load index (PLI) to comprehensively evaluate the pollution levels of heavy metal elements in topsoil. Among them, the single pollution index exhibits the pollution degree of each heavy metal, which is defined as the following:(1)$$Pi=\frac{C_{i}}{S_{i}}$$
where *C_i_* and *S_i_* are the concentrations of the *i*th heavy metal in the soil sample and its background concentration in soil, respectively [16]. According to a previous study [17], pollution exists when *P_i_* > 1 and can be categorized as the moderate contamination factor (1 ≤ *P_i_* < 3), the considerable contamination factor (3 ≤ *P_i_* < 6), the high contamination factor (*P_i_* ≥ 6), and the low contamination factor (*P_i_* < 1).

The *PN* is a multi-factor environmental quality index that takes into account the extreme values, especially considering the most polluting factors. *PN* describes the integrated pollution level in the study area and is calculated as follows [18,19]:(2)$$PN=\sqrt{\frac{{\left(Pi\right)}_{\max}^{2}+{\left(Pi\right)}_{mean}^{2}}{2}}$$
where *P_i(_*_max)_ and *P_i(m_**_e_**_an)_* are the maximum and average values of *P_i_* for all target heavy metal elements, respectively. The degree of contamination of heavy metals is classified according to the following criteria: safe (*PN* ≤ 0.7), warning (0.7 < *PN* ≤ 1), light pollution (1 < *PN* ≤ 2), moderate pollution (2 < *PN* ≤ 3), and heavy pollution (*PN* > 3).

*PLI* is the load capacity of a pollutant for a region or an environmental element, and it is usually used to evaluate the pollution degree of all heavy metals in soil, which is calculated according to the following equation [20,21].
(3)$$PLI=(\frac{C_{1}}{S_{1}}\times \frac{C_{2}}{S_{2}}\times \cdots \times {\frac{C_{i}}{S_{i}})}^{\frac{1}{n}}.$$

Four categories are recognized for *PLI*: the low level of pollution (*PLI* ≤ 1), the moderate level of pollution (1 < *PLI* ≤ 2), the high level of pollution (2 < *PLI* ≤ 5), and the extremely high level of pollution (*PLI* > 5).

*I_geo_* was proposed by Müller (1969) and shows the pollution intensity of individual heavy metal elements according to the following equation [22,23], and this method not only reflects the natural variation characteristics of heavy metal distribution, but also can discern the impact of anthropogenic activities on the environment, and is an important parameter to distinguish the impact of anthropogenic activities:(4)$$I_{geo}=Log_{2}\frac{Ci}{1.5\times S_{i}}.$$

*I_geo_* is classified as uncontaminated (*I_geo_* ≤ 0), uncontaminated to moderately contaminated (0 ˂ *I_geo_* ≤ 1), moderately contaminated (1 ˂ *I_geo_* ≤ 2), moderately to strongly contaminated (2 ˂ *I_geo_* ≤ 3), strongly contaminated (3˂ *I_geo_* ≤ 4), strongly to extremely contaminated (4 ˂ *I_geo_* ≤ 5), and extremely contaminated (*I_geo_* > 5).

### 2.4. Ecological Risks of Soil Heavy Metal

The potential ecological risk index method (RI) was proposed in 1980 by Hakanson Lars, a Swedish scientist [24]. This method is used to assess the potential risk for the ecological environment by heavy metals in soil/sediment [25]. The calculated equation of PERI is referenced in previous studies [12,13,25] and is defined as the following:(5)$$E_{r}^{i}=T_{x}^{i}\times \frac{C_{i}}{S_{i}},$$
(6)$$\mathrm{RI}=\sum_{i=1}^{n}E_{r}^{i},$$
where *T_x_^i^* signifies the toxic response coefficient of each heavy metal, and the value is defined as Cd = 30, Hg = 40, As = 10, Cu = Pb = Ni = 5, Cr = 2, and Zn = 1 [24,26]. $E_{r}^{i}$ and RI are the single potential ecological risk index and the integrated potential ecological risk index, respectively [26]. The ecological risk analysis standard of the potential ecological risk assessment is shown in Appendix A.

### 2.5. Health Risk of Heavy Metals

Exposure models for different contamination pathways can be used to quantitatively describe the effects of soil contaminants on local human health. In this study, an exposure assessment model for non-sensitive plots was used [27], and its thresholds can be used to assess the potential health risks of heavy metals to adults living and working near landfills. The three main exposure pathways were considered: ingestion, dermal contact, and inhalation [28,29]. The evaluation model is as follows:(7)$${\mathrm{HQ}}_{\mathrm{ingest}}=\frac{C_{i}\times {OSIR}_{a}\times {ED}_{a}\times {EF}_{a}\times {ABS}_{o}}{{BW}_{a}\times {AT}_{nc}\times {RfD}_{o}\times SAF}\times 10^{-6},$$
(8)$${\mathrm{HQ}}_{\mathrm{dermal}}=\frac{C_{i}\times {SSAR}_{a}\times {ED}_{a}\times {EF}_{a}\times E_{v}\times {ABS}_{d}\times {SAE}_{a}}{{BW}_{a}\times {AT}_{\mathrm{nc}}\times {RfD}_{d}\times SAF}\times 10^{-6},$$
(9)$${\mathrm{HQ}}_{\mathrm{inhale}}=\frac{C_{i}\times {PM}_{10}\times {DAIR}_{a}\times PIAF\times {ED}_{a}\times (fspo\times {EFO}_{a}+fspi\times {EFI}_{a})}{{BW}_{a}\times {AT}_{nc}\times {RfD}_{i}\times SAF}\times 10^{-6},$$
(10)$${CR}_{\mathrm{ingest}}=\frac{C_{i}\times {OSIR}_{a}\times {ED}_{a}\times {EF}_{a}\times {ABS}_{o}\times {SF}_{o}}{{BW}_{a}\times {AT}_{ca}}\times 10^{-6},$$
(11)$${CR}_{\mathrm{dermal}}=\frac{C_{i}\times {SAE}_{a}\times {SSAR}_{a}\times {ED}_{a}\times {EF}_{a}\times E_{v}\times {ABS}_{d}\times {SF}_{d}}{{BW}_{a}\times {AT}_{ca}}\times 10^{-6},$$
(12)$${CR}_{\mathrm{inhale}}=\frac{C_{i}\times {SF}_{i}\times {PM}_{10}\times {DAIR}_{a}\times PIAF\times {ED}_{a}\times (fspo\times {EFO}_{a}+fspi\times {EFI}_{a})}{{BW}_{a}\times {AT}_{ca}}\times 10^{-6},$$
(13)$$CR={CR}_{\mathrm{ingest}}+{CR}_{dermal}+{CR}_{\mathrm{inhale}},$$
(14)$$HQ={HQ}_{\mathrm{ingest}}+{HQ}_{dermal}+{HQ}_{\mathrm{inhale}},$$
where *HQ_ingest_*, *HQ_dermal_*, and *HQ_lnhale_* represent the hazard quotient of individual soil heavy metals in the exposure pathways of ingestion, dermal contact, and inhalation, and *CR_ingest_*, *CR_dermal_*, and *CR_lnhale_* are the carcinogenic risk of the three exposure pathways, respectively. *C_i_* shows the concentration of each heavy metal element in topsoil (mg/kg); other exposure parameters are shown in Appendix A.

### 2.6. Source Apportionment of Soil Heavy Metals

The inter-element relationships can provide information on the source and pathways of heavy metals [30]. Significant positive correlations between heavy metals suggest that they may have similar sources or geochemical characteristics, and can help identify the source of elements [31,32]. The principal component analysis (PCA) is a mathematical process that uses orthogonal transformations to convert potentially correlated variables into linearly uncorrelated ones. In this paper, SPSS 19.0 software (International Business Machines Corporation, Armonk, NY, USA) was used to determine the sources of heavy metals using correlation analysis and principal component analysis.

## 3. Results and Discussion

### 3.1. Soil Properties and Heavy Metal Concentrations

According to the soil texture analysis, the soil types in the study area belong to sandy clay and clayey silt. The soil pH ranged from 7.54 to 10.90, with an average value of 8.59 ± 0.64. Heavy metal concentrations of topsoil samples within 0.5 km of MSW landfill sites are listed in Figure 2. The average concentrations of Cu, Pb, Zn, Cr, Ni, Cd, As, and Hg were 29.96 ± 16.10, 36.43 ± 13.70, 84.93 ± 49.49, 54.60 ± 10.42, 25.27 ± 5.57, 0.39 ± 0.92, 26.97 ± 4.89, and 0.043 ± 0.054 mg/kg, respectively. From the characteristics of heavy metal concentrations in different depths of soil layers, the concentration of heavy metals in topsoil is relatively high, and the concentration characteristics of deep soil layers are similar between different depths. The content of heavy metals in the soil around the landfill decreases one by one as the relative distance to the landfill increases. Due to the accumulation of heavy metal content in the soil, individual monitoring sites showed abnormal conditions (Figure 3), and this result is consistent with the relevant results found in the literature [33]. In terms of coefficients of variation, Cd and Hg are more variable and subject to greater anthropogenic influence.

Among all sampling sites, the sampling site concentrations of 69% (target metal: Cu), 64% (target metal: Pb), 68% (target metal: Zn), 90% (target metal: Cr), 76% (target metal: Ni), 79% (target metal: Cd), and 94% (target metal: As) exceeded the background concentrations of soil heavy metals in Lhasa City, while the average concentration of Hg was lower than the background value. However, according to the China Soil Environmental Quality Standard (GB15618-1995), the heavy metal concentrations of about 81.99% (target metal: Cu), 41.7% (target metal: Pb), 87.5% (target metal: Zn), 100% (target metal: Cr), 98.6% (target metal: Ni), 75.0% (target metal: Cd), 1.4% (target metal: As), and 95.8% (target metal: Hg) were evaluated as level I. Accordingly, the concentrations of about 33.3% (target metal: Pb), 11.1% (target metal: Zn), 1.4% (target metal: Ni), 4.2% (target metal: Cd), 33.3% (target metal: As), and 4.2% (target metal: Hg) of sampling sites were evaluated as level Ⅱ, while those of about 1.4% (target metal: Zn), 6.9% (target metal: Cd), and 65.3% (target metal: As) of sampling sites were evaluated as level Ⅲ in Table 1. In the future, Cu, Pb, As, Zn, and Cd will be the main pollution factors that should attract critical concern in landfills, which has been reported previously [34].

Compared with landfill soil heavy metals in Guangzhou, Shanghai, Guiyang, and Shandong, the overall soil heavy metal content of landfill in Lhasa is lower than that of landfill soil in the mentioned cities. Compared with previous studies [14,39] conducted in the Qinghai-Tibet Plateau, the soil heavy metal concentrations analyzed in this study were lower in Table 1. The main reason may be closely related to the domestic waste component of Lhasa City. Dan et al. [40] reported that the physical components of MSW in Tibet are dominated by paper (23.74%), inert waste (22.83%), kitchen waste (20.45%), and plastic (14.84%), followed by wood (6.23%), metal (5.12%), glass (4.73%), textiles, and leather (4.5%). Generally, heavy metals mainly originate from electronic products, industrial solid waste, domestic waste incineration, etc. [41,42,43]. The ratio of metal was relatively low compared to other MSW components.

The distribution of individual heavy metals shows elemental specificity and sharp spatial variation (Figure 4). The average concentrations of most heavy metals in the soils of this study area were generally higher than the corresponding background values. The average value of *P_i_* for each heavy metal decreased in the order Cd (1.42) > As (1.35) > Cu (1.32) > Cr (1.29) > Zn (1.26) > Ni (1.20) > Pb (1.15) > Hg (0.43). The *P_i_* values of Cd, As, Cu, Cr, Zn, Ni, and Pb were in the range of 1–2, which was classified as light contamination. Hg, with *P_i_* ≤ 1, was classified as low contamination in the study area. The pollution levels of soil heavy metals in different sampling sites are shown in Figure 4. Soil heavy metal contamination around the landfill is mainly concentrated in the southeast area, and there is no obvious pattern of sampling sites in other directions. The pollution degrees of 65% (target metal: Cu), 63% (target metal: Pb), 65% (target metal: Zn), 90% (target metal: Cr), 76% (target metal: Ni), 75% (target metal: Cd), 94% (target metal: As), and 6% (target metal: Hg) of sampling sites were classified as moderate pollution. Approximately 4% (target metal: Cu), 1% (target metal: Pb), 3% (target metal: Zn), and 4% (target metal: Cd) of sampling sites were classified as considerable pollution, and other sampling sites were classified as low pollution. According to the field investigation, the major reason might be the waste disposal activities, which are located in the east and south of the landfill site, such as a hazardous waste treatment center, domestic waste incineration, and leachate treatment station. Related research shows that various reasons, such as vehicle emissions, can cause the lead content in the soil on both sides of the road to exceed the standard [44]. The concentration of heavy metals Cd, As, and Zn in the soil around the leachate treatment station will increase due to its influence [45]. To some extent, domestic waste incineration plants can also lead to heavy metal (Pb, Cu, Zn, Cd, Hg, and As) contamination of the surrounding soil [46], which is similar to the study area environment.

### 3.2. Soil Heavy Metal Contamination Risk

Based on the background values, three methods (*PN*, *PLI*, *I_geo_*) were adopted to systematically evaluate the soil pollution around the landfill. The *PN* values of heavy elements in the soil were not as high as expected, ranging from 0.74 to 3.83, while the mean value was 1.51. A number of 3, 6, 60, and 3 sampling sites showed high pollution, moderate pollution, light pollution, and warning, respectively. The sampling areas of NE, NW, SW, N, S, W, and E showed light pollution, while the SE area exhibited moderate pollution. The pollution degrees of heavy metals in different sampling directions are ranked in the order SE > NE > NW > SW > N > S > W > E (Figure 5). Heavy metal elements, including Cu, Zn, and Cd, are the main pollution source for *PN* in the southeast, with the classification of moderate contamination at 11.1%, 33.3%, and 11.1%, and heavy contamination percentages at 33.3%, 22.2%, and 22.2%, respectively.

*PLI* also provides comparative information for assessing heavy metal contamination. *PLI* values for all sampling sites ranged from 0.55 for the eastern sampling sites to 2.24 for the southeastern sampling sites, with a mean value of 1.07. Based on the *PLI* ranking criterion, 1, 39, and 32 of 72 sampling sites showed high pollution, moderate pollution, and low pollution levels, respectively. Among all of the sample sites, the pollution levels of the sampling sites in the southeast were relatively high. Cu and Zn served as the dominant contributors of *PLI* and were assessed as having a high pollution level, and other heavy metals were evaluated as having a moderate or low pollution level. The comprehensive pollution levels of sampling sites N, NE, NW, SE, SW were classified as having moderate pollution, and other sampling sites are considered as having a low level of pollution. The order of pollution degree of individual sampling sites in the study area is ranked as: SE > NE > NW > SW > N > S > W > E (Figure 5).

Although the two evaluation methods focus on different perspectives, the evaluation results are similar. The topsoil surrounding the landfill site is polluted by heavy metals and most of them are classified into moderate and light pollution levels. The comprehensive pollution level of the southeast area is significantly higher than other sampling areas, probably due to the intensive waste disposal activities. The main pollution factors are As, Cd, and Cr in the study area.

Average *I_geo_* values of all the heavy metals at the sampling sites ranged from −3.07 to 2.07 (Hg and Cd), while the minimal and maximal *I_geo_* values were −4.24 (Hg) and 4.87 (Cd), respectively. According to *I_geo_* grading criteria [22,23], approximately 80.56% of the sampling sites were uncontaminated with Cu, while uncontaminated–moderate contamination effects were found in 19.44% of the sites. Approximately 90.28% of the sampling sites were not contaminated with heavy metal Pb, while 9.72% of the sites were between uncontaminated and moderately contaminated with Pb. Approximately 87.52% of the sampled sites were uncontaminated and 12.5% of the sampled sites were between uncontaminated and moderately contaminated with Zn. Cr was not found in 75.0% of the sampled sites, while in 25.0% of the sampled sites, Cr was between uncontaminated and moderately contaminated. Ni had an uncontaminated level in 86.11% of sampling sites while it had an uncontaminated–moderately contaminated level in 13.89% of sites. For Cd, 66.67% of the sampling sites were uncontaminated and 19.44% of the sampling sites were contaminated with a level between uncontaminated and moderately contaminated. Approximately 79.17% of the sampling sites were not contaminated with As, and 20.83% of the sampling sites had an uncontaminated–moderate pollution level. Approximately 95.83% of the sampling sites were uncontaminated with Hg, and 4.17% of the sampling sites were uncontaminated–moderately contaminated with Hg. Approximately 88% of the sampling sites were classified as uncontaminated. However, the soil heavy metals in the E, S, W, N, NE, NW, SE, and SW sampling sites of the study area showed a similar trend in which the average *I_geo_* values were less than zero, and classified as uncontaminated (Figure 6). The *I_geo_* values of Cu, Pb, Zn, Cr, and Cd were significantly varied in the southeast landfill sites. Among them, the metals Cu, Pb, Zn, and Cr exhibited similar patterns to the uncontaminated and moderately contaminated samples, while Cd indicated a moderate to strong contamination. The average *I_geo_* values of individual heavy metals in the study area were less than 1, showing that they were uncontaminated to moderately contaminated, and the degree of pollution is ranked as Cd > As > Cr > Cu > Zn > Ni > Pb > Hg (Figure 6).

### 3.3. Soil Ecological Risk

The evaluation results of potential ecological risks in the study area are shown in Figure 7. RI values of heavy metal elements in soil ranged from 44.5 (E) to 318.4 (SE) and the average value of RI was 95.7. Based on the RI ranking criterion, one out of the 72 sampling sites exhibited considerable ecological risk, while five sites showed moderate ecological risk. In contrast, approximately 66 sites exhibited low ecological risk. Cd, As, and Hg served as the dominant contributors of ecological risk in the study area and contributed over 45%, 14.1%, and 18%, respectively. The order of ecological risk of different sampling sites is ranked as: SE (156.23) > NW (100.56) > NE (99.45) > N (96.81) > SW (96.57) > S (75.20) > E (70.74) > W (70.25). Among them, only the southeast sampling sites in the study area showed a moderate ecological risk.

The ecological risk of single heavy metal elements was identified as a low ecological hazard with the mean E_r_^i^ ranging from 1.26 (Zn) to 42.72 (Cd), while only the heavy metal Cd was assessed as moderate ecological risk (40 ≤ E_r_^i^ (42.72) ˂ 80). Twenty-five and five sites of the 72 sampling sites exhibited a moderate and heavy ecological hazard (target metal: Cd), respectively. Two and two sites out of the 72 sampling sites exhibited a moderate and heavy ecological hazard (target metal: Hg), respectively. The other six soil heavy metals (Cu, Pb, Zn, Cr, Ni, and As) showed low ecological hazards in 72 sampling sites. In summary, the comprehensive potential ecological risk of the study area is relatively low. Cd, As, and Hg are the main pollution factors of the ecological risk. The ecological risk of the southeast (SE) sampling area in the landfill site was relatively high and should be the main focus of future studies and corresponding protection measures should be taken.

### 3.4. Health Risk Assessment of Soil Heavy Metals

This article only evaluates the health risks of adults because the location of the research area is considered non-sensitive. The carcinogenic risks and non-carcinogenic risks posed by heavy metals (Cd, Cr, As, Hg, Pb, Cu, Zn, and Ni) in soils surrounding the landfill site in Tibet for adults through different exposure pathways (inhalation, dermal contact, ingestion) were evaluated.

#### 3.4.1. Carcinogenic Risk

Carcinogenic risks (CRs) of Cd and As were estimated, and other heavy metals were neglected because of the lack of slope factors. Through different exposure pathways, the CRs of Cd and As for adults were in the ranges of 2.73 × 10^−11^ to 1.47 × 10^−5^ and 8.04 × 10^−10^ to 2.39 × 10^−5^, respectively, and the CRs through pathways of ingestion, dermal contact, and inhalation are in the following order: *CR_ingest_* > *CR_inhale_* > *CR_dermal_*. Ingestion is the dominant pathway for *CR*, with contribution rates of 86.8% and 98.6% for the *CR* of As and Cd, respectively. As shown in Table 2, the *CR_ingest_* and *CR* of As are 1.50 × 10^−5^ and 1.73 × 10^−5^, respectively, exceeding the acceptable goal of 1.0 × 10^−6^ [27]. Therefore, people living in the vicinity of landfill sites are at high cancer risk from As exposure, which should be controlled in the future.

#### 3.4.2. Non-Carcinogenic Risk

The non-carcinogenic risk of individual heavy metals can be expressed as a hazard quotient (HQ). The accepted standard of the HQ is 1.0, and there is a significant health hazard when the HQ exceeds 1 [47,48]. The hazard index (HI) is the sum of the hazard quotient of a single element in multiple exposure pathways. HI values > 1 suggest that there will be adverse health effects.

The non-carcinogenic risks to adults posed by soil heavy metals through different pathways are shown in the Table 3. Through different pathways, the HQ for adults of each heavy metal element ranged from 5.25 × 10^−8^ to 8.01 × 10^−1^. Ingestion is the dominant exposure pathway for hazard quotients of each element, with an average contribution rate of 51.75%. The total hazard indices of each heavy metal are ranked as: As > Cr > Ni > Pb > Cu > Cd > Zn > Hg. As and Cr were the main sources of non-carcinogenic risks, with average contributions of 69.12% and 18.35%, respectively. The maximum HI value of all heavy metals was 0.983 (lower than 1), indicating no significant non-carcinogenic health risk for adults living around the landfill sites.

### 3.5. Possible Sources of Soil Heavy Metals

The Pearson’s correlation matrix between heavy metals is shown in Table 4. Cu, Pb, Zn, Hg, and Cd were significantly positively correlated with each other (*p* ˂ 0.01). Ni was significantly positively correlated with Cr (*p* ˂ 0.01) and Cr was positively correlated with Cu and Pb (*p* ˂ 0.05), indicating the same sources in each group. As was not correlated with other heavy metals.

For PCA analysis, the result of the KMO (0.779) and Bartlett’s test (*p* ˂ 0.001) indicated good coordination with PCA. Three principal components (PCs) were extracted, which explain 92.22% of the total variance (among them, PC1: 55.52%; PC2: 23.72%; PC3: 12.98). Consistent with the results of the correlation analysis, PC1 was dominated by Cu, Pb, Zn, Cd, and Hg; PC2 was dominated by Cr and Ni; and PC3 was heavily composed of As. The average concentrations of Cu, Pb, Zn, and Cd in topsoil were higher than their background values. The coefficients of variance (CVs) of these metals were relatively high (>0.3) (Table 5), indicating that these heavy metal elements were mainly affected by anthropogenic sources [31,49]. Meanwhile, the concentrations of Cu, Pb, Zn, Cd, and Hg for the SE orientation were significantly higher than those at other sites (Figure 4). The intensive waste disposal activities including hazardous waste treatment, leachate treatment, MSW incineration, and dumping of construction waste located in the SE part of the site were considered the main source of Cu, Pb, Zn, Cd, and Hg in soil. PC1 was identified as intensive waste disposal activities. Ni and Cr are generally regarded as the marker elements of natural sources. These two elements are controlled by the weathering of the parent material and pedogenesis [50,51]. Therefore, PC2 was heavily loaded by Ni and Cr, thus Ni was regarded as a natural source. The accumulation of As in the soil at the study area probably resulted from the construction and operation of the landfill Phase II Project in Lhasa. PC3 might be attributed to construction activities. Interestingly, Cr was heavily loaded in PC2 and moderately loaded in PC1. Cr was comprehensively affected by both natural sources and intensive waste disposal activities.

## 4. Conclusions

This article adopts the single pollution index (*P_i_*), the Nemerow pollution index (*PN*), the geo-accumulation index (*I_geo_*), and the pollution load index (*PLI*) to comprehensively evaluate the pollution levels of heavy metal elements in topsoil at the MSW landfill close to Lhasa, Tibet. Soil heavy metal contamination of the study area is relatively low. The maximal concentrations of Cd, As, Cu, Cr, Zn, Ni, and Pb were 1–2 times higher than the corresponding background value, which was classified as light contamination. Based on the *PN* sampling, 3, 6, and 60 sites showed high, moderate, and light pollution status, respectively. According to PLI, 1, 39, and 32 sampling sites possessed high, moderate, and low contamination levels. Cu, Cd, and Zn served as the dominant *PLI* and *PN* contributors. *I_geo_* values of all the heavy metals in the sampling sites ranged from −3.07 to 2.07 (Hg and Cd), while the minimal and maximal *I_geo_* values were −4.24 (Hg) and 4.87 (Cd), respectively. Eighty-eight percent of the sampling sites were classified as uncontaminated. The potential ecological risk of the study area was low, as one out of the 72 sampling sites exhibited considerable ecological risk, Cd, As, and Hg were the main pollution factors in ecological risk. The carcinogenic risk of soil heavy metals for adults was 1.73 × 10^−5^, exceeding the acceptable level according to the risk control values of the US Environmental Protection Agency. There were no significant non-carcinogenic health risks for adults. Arsenic (As) is the most contributively toxic element to human health risk. Three main potential sources for soil heavy metals were identified: intensive waste disposal activities, natural sources, and construction activities. Due to the waste disposal activities, the pollution levels and ecological risks of soil heavy metals in the southeast part of the landfill site were higher. The heavy metal concentrations of the soil surrounding the landfill site in the neighboring area of Lhasa were significantly affected by anthropogenic activities. Therefore, more efficient environmental protection measures are urgently needed.

## Figures and Tables

**Figure 1 ijerph-19-10704-f001:**
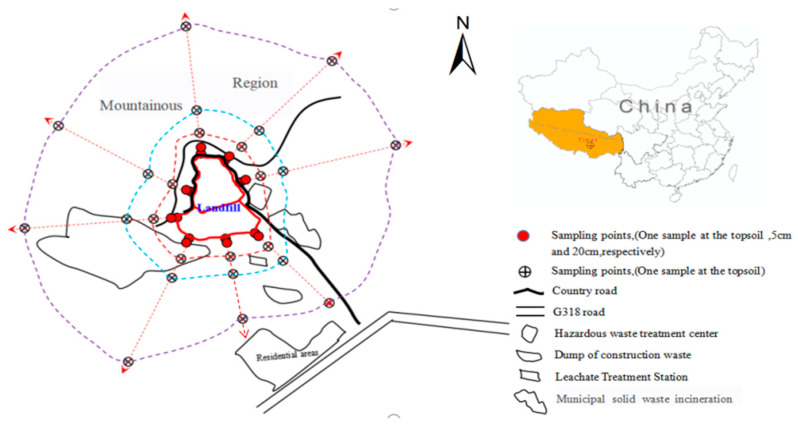
Location of the sampling sites around the domestic waste sanitary landfill site.

**Figure 2 ijerph-19-10704-f002:**
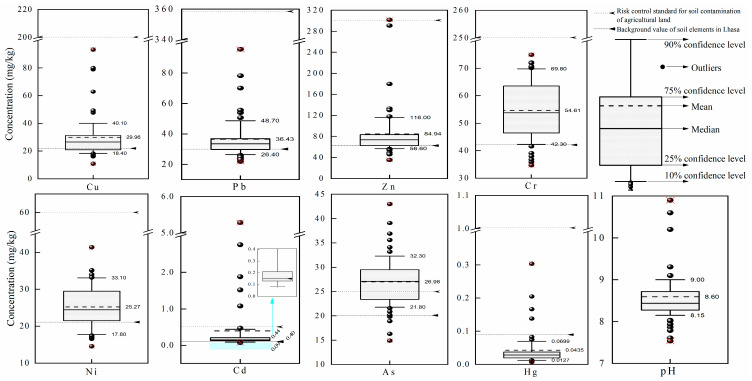
The soil heavy metal concentration characteristics. ×: represents maximum and minimum values of soil heavy metal concentrations.

**Figure 3 ijerph-19-10704-f003:**
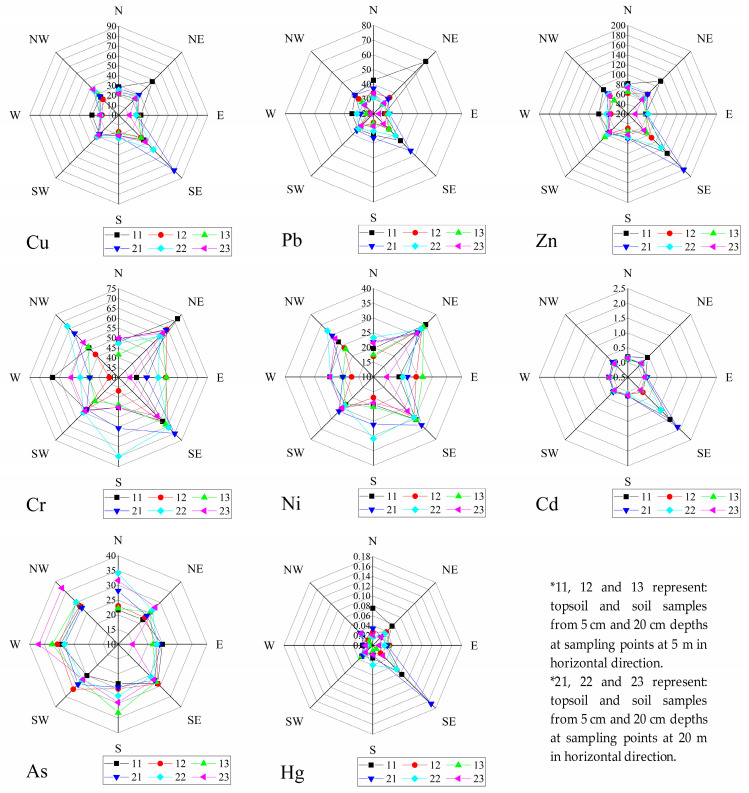
Characteristics of heavy metal concentrations in soil at different depths.

**Figure 4 ijerph-19-10704-f004:**
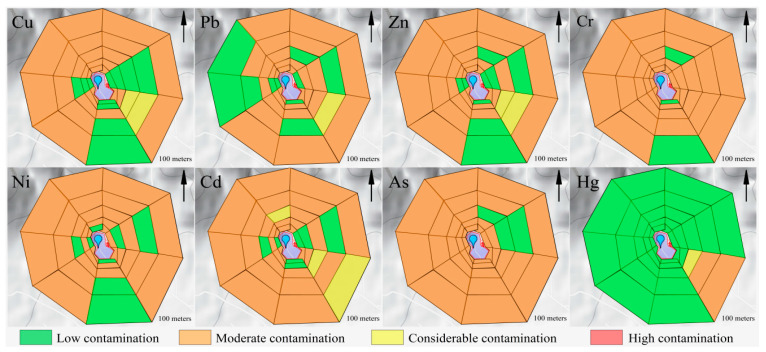
Distribution of soil heavy metal pollution characteristics.

**Figure 5 ijerph-19-10704-f005:**
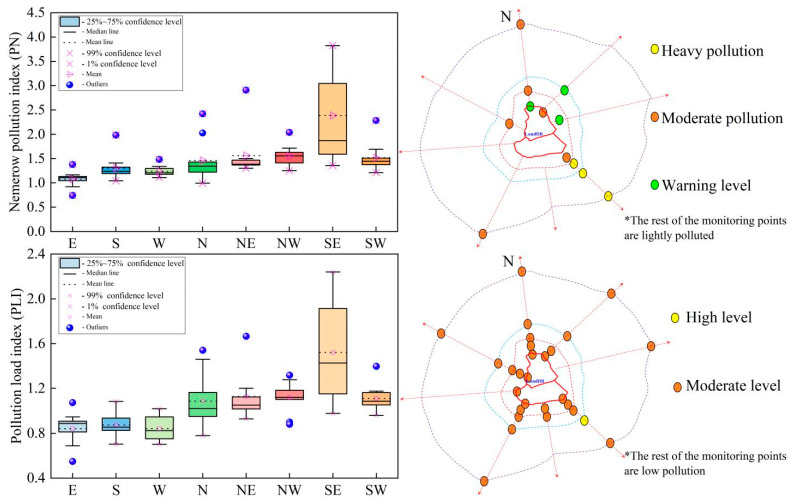
Levels of heavy metal contamination in landfill soil.

**Figure 6 ijerph-19-10704-f006:**
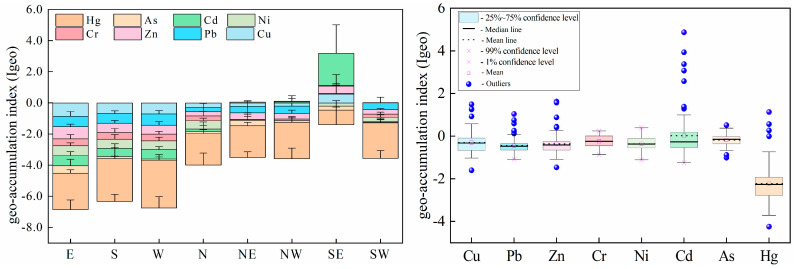
Characteristics of *I_geo_* risk values at landfill sites.

**Figure 7 ijerph-19-10704-f007:**
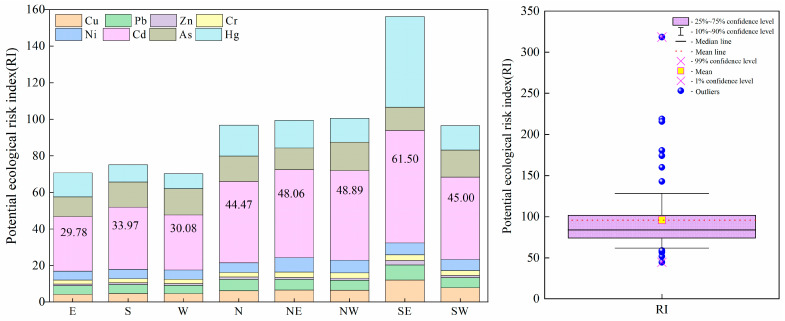
Ecological risk characterization of soil heavy metals in landfill.

**Table 1 ijerph-19-10704-t001:** Summary statistics of heavy metal concentrations (mg/kg) in the landfill site.

	Cu	Pb	Zn	Cr	Ni	Cd	As	Hg	PH
Arithmetic mean	29.96	36.43	84.94	54.61	25.27	0.40	26.98	0.04	8.60
Standard deviation	16.10	13.71	49.49	10.42	5.57	0.92	4.89	0.05	0.64
Coefficient of variance	0.54	0.38	0.58	0.19	0.22	2.33	0.18	1.26	0.07
Skewness	2.61	2.90	3.50	0.10	0.34	4.57	0.49	3.63	2.19
Kurtosis	7.20	9.46	12.87	−0.82	−0.34	21.50	1.34	14.02	5.65
Guangzhou [35]	Null	31.6	60.1	77.1	20.5	0.072	Null	0.091	Null
Shanghai [36]	68.1	94.5	1437.8	69.9	34.1	4.57	356.7	0.043	Null
Guiyang [37]	Null	49.72	1025.4	Null	122.96	13.67	Null	Null	Null
Shandong [38]	128.48	333.34	301.95	108.86	30.23	0.64	18.2	0.045	Null
Background value of soil elements in Lhasa City [16]	22.00	31.00	65.00	42.00	21.00	0.12	20.00	0.092	8.20
Chinese soil guidelines (Grade II) (CEPA, 1995)	200.00	350.00	300.00	250.00	60.00	0.60	25.00	1.00	Null

**Table 2 ijerph-19-10704-t002:** Cancer risk of Cd and As by three exposure routes from the soil around the landfill.

Carcinogenic Risk	*CR_ingest_*			*CR_dermal_*			*CR_inhale_*			*CR*
	Avg	Max	Min	Avg	Max	Min	Avg	Max	Min	
Cd	9.13 × 10^−7^	1.47 × 10^−5^	2.12 × 10^−7^	1.18 × 10^−10^	1.90 × 10^−9^	2.73 × 10^−11^	1.14 × 10^−8^	1.85 × 10^−7^	2.65 × 10^−9^	9.25 × 10^−7^
Contribution	98.67%			0.09%			1.24%			-
As	1.50 × 10^−5^	2.39 × 10^−5^	8.31 × 10^−6^	1.45 × 10^−9^	2.32 × 10^−9^	8.04 × 10^−10^	2.25 × 10^−6^	3.59 × 10^−6^	1.24 × 10^−6^	1.73 × 10^−5^
Contribution	86.84%			0.17%			12.99%			-

**Table 3 ijerph-19-10704-t003:** Non-carcinogenic risk of heavy metals in topsoil around the landfill site.

Non-Carcinogenic Risk	*HQ_ingest_*			*HQ_inhale_*			*HQ_dermal_*			HI
	Max	Min	Avg	Max	Min	Avg	Max	Min	Avg	
Cu	1.25 × 10^−2^	1.46 × 10^−3^	3.89 × 10^−3^	6.51 × 10^−2^	7.60 × 10^−3^	2.03 × 10^−2^	4.03 × 10^−5^	4.71 × 10^−6^	1.26 × 10^−5^	2.42 × 10^−2^
Pb	1.46 × 10^−1^	3.35 × 10^−2^	5.44 × 10^−2^	1.80 × 10^−3^	4.20 × 10^−4^	6.80 × 10^−4^	4.70 × 10^−4^	1.10 × 10^−4^	1.80 × 10^−4^	5.53 × 10^−2^
Zn	5.39 × 10^−3^	6.30 × 10^−4^	1.46 × 10^−3^	6.75 × 10^−5^	7.88 × 10^−6^	1.83 × 10^−5^	1.74 × 10^−5^	2.03 × 10^−6^	4.72 × 10^−6^	1.48 × 10^−3^
Cr	1.34 × 10^−1^	6.19 × 10^−2^	9.69 × 10^−2^	2.09 × 10^−1^	9.69 × 10^−2^	1.52 × 10^−1^	1.73 × 10^−2^	7.90 × 10^−3^	1.25 × 10^−2^	2.61 × 10^−1^
Ni	1.11 × 10^−2^	3.91 × 10^−3^	6.73 × 10^−3^	1.29 × 10^−1^	4.53 × 10^−2^	7.81 × 10^−2^	8.90 × 10^−4^	3.20 × 10^−4^	5.40 × 10^−4^	8.54 × 10^−2^
Cd	2.83 × 10^−2^	4.07 × 10^−4^	1.75 × 10^−3^	1.48 × 10^−1^	2.12 × 10^−3^	9.20 × 10^−3^	3.66 × 10^−6^	5.25 × 10^−8^	2.26 × 10^−7^	1.10 × 10^−2^
As	7.67 × 10^−1^	2.66 × 10^−1^	4.81 × 10^−1^	8.01 × 10^−1^	2.77 × 10^−1^	5.02 × 10^−1^	7.43 × 10^−5^	2.57 × 10^−5^	4.66 × 10^−5^	9.83 × 10^−1^
Hg	5.42 × 10^−3^	1.30 × 10^−4^	7.11 × 10^−4^	2.80 × 10^−4^	6.79 × 10^−6^	3.71 × 10^−5^	2.50 × 10^−4^	6.01 × 10^−6^	3.28 × 10^−5^	7.81 × 10^−4^
Contribution	51.75%			46.94%			1.31%			-

**Table 4 ijerph-19-10704-t004:** Pearson’s correlation matrix between heavy metals (*n* = 8).

Heavy Metals	Cu	Pb	Zn	Cr	Ni	Cd	As	Hg
Cu	1.000							
Pb	0.933 **	1.000						
Zn	0.958 **	0.955 **	1.000					
Cr	0.812 *	0.758 *	0.705	1.000				
Ni	0.497	0.423	0.332	0.899 **	1.000			
Cd	0.901 **	0.894 **	0.974 **	0.634	0.263	1.000		
As	0.038	−0.095	−0.033	0.049	0.031	−0.169	1.000	
Hg	0.906 **	0.936 **	0.975 **	0.632	0.252	0.982 **	−0.211	1.000

(**: *p* ˂ 0.01; *: *p* ˂ 0.05).

**Table 5 ijerph-19-10704-t005:** Rotated component matrix of heavy metals in soils.

Factors	Cu	Pb	Zn	Cr	Ni	Cd	As	Hg	% of Variance	Cumulative %
PC1	0.89	0.88	0.97	0.39	0.02	0.94	−0.11	0.95	55.52	55.52
PC2	0.31	0.26	0.16	0.88	0.96	0.01	0.12	0.07	23.72	79.24
PC3	0.11	−0.14	−0.05	0.05	0.1	−0.1	0.98	−0.11	12.98	92.22

## Data Availability

The data used to support the findings of this study are available from the corresponding author.

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
