# Peer review of "Pollution Levels and Risk Assessment of Heavy Metals in the Soil of a Landfill Site: A Case Study in Lhasa, Tibet"

_ijerph, 2022, doi:10.3390/ijerph191710704_

Round 1
Reviewer 1 Report
This paper describes a Pollution levels and risk assessment of heavy metals in the soil of a landfill site: A case study in Lhasa, Tibet. It seems interesting. I think it can be consider to publish if the following issues are solved:
1. some recent achievements should be summarize in abstract.
2. The preparation methods also should be discussed in detail
3, some recent revalent papers should be cited: Small Methods 2200314 (2022) doi:10.1002/smtd.202200314; Colloids and Surfaces A: Physicochemical and Engineering Aspects 646, 128962.
5. the english level should be improved.
Author Response
Dear Reviewer,
This Article was modified and improved as follows:
- some recent achievements should be summarize in abstract.
Revised and improved
- The preparation methods also should be discussed in detail
Revised and improved(The methods have be discussed in detail )
3, some recent revalent papers should be cited: Small Methods 2200314 (2022) doi:10.1002/smtd.202200314; Colloids and Surfaces A: Physicochemical and Engineering Aspects 646, 128962.
Revised and improved (Added to the article at the appropriate content)
- the english level should be improved.
Modified and improved
Kind regards,
Mr. Peng,

Reviewer 2 Report
Any landfill or waste disposal need monitoring, once more they are near inhabited areas or have potential influence on them. Even if this subject is not new in terms of measurement methodology and sample collection principles, the method of interpretation exposed in this paper allows obtaining reliable and possibly repeatable results (depending on the sampling points). The parameters quantified in this way are easier to interpret graphically, on defined areas.
In few places, the general information is slightly confused or contradictory (see the abstract, the second and third sentences – referring to the same area, Tibet, or the second sentence from chapter 2.2). I have marked also the sentences without a verb. Some graphics are too small to be clear (Figure 3). It was observed also a recurrent different writing of some indicators/symbols (e.g Igeo / Igeo) in the body of paper, being recommended the consistency / uniformity.
But the research was well conducted and it was paid attention to the presentation of the research frame and to the interpreting of the results according to the chosen methodology. The references for the limits of heavy metals pollutants were clear. The presentation and the approach to the subject respect the scientific rules, and the results are clearly stated.
The typing and grammatically suggestions were made directly on the text.
I consider it is a useful research and interesting paper related to the environment pollution and protection measures in the vicinity of Lhasa city, presenting a great importance for the area of ​​Tibet.

Author Response
Dear Reviewer,
Based on the valuable suggestions you have provided, I have revised and perfected each of them. Thank you very much for your patience in giving valuable guidance and advice on this article, Please see the attachment.
Kind regards,
Mr. Zhou,

This manuscript is a resubmission of an earlier submission. The following is a list of the peer review reports and author responses from that submission.